# Copper-Content Dependent Structural and Electrical Properties of CZTS Films Formed by "Green" Colloidal Nanocrystals

Volodymyr Dzhagan [1,2,*], Oleksandr Selyshchev [3,4], Serhiy Kondratenko [2], Nazar Mazur [1],
Yevhenii Havryliuk [1,3,4], Oleksandra Raievska [3,4,5], Oleksandr Stroyuk [5] and Dietrich R. T. Zahn [3,4]

[1]  V. Lashkaryov Institute of Semiconductors Physics, National Academy of Sciences of Ukraine, 41 Nauky Av.,
     03028 Kyiv, Ukraine; nazarmazur@isp.kiev.ua (N.M.); yevhenii.havryliuk@physik.tu-chemnitz.de (Y.H.)
[2]  Physics Department, Taras Shevchenko National University of Kyiv, 64 Volodymyrs'ka St.,
     01601 Kyiv, Ukraine; kondratenko@ukr.net
[3]  Semiconductor Physics, Chemnitz University of Technology, 09107 Chemnitz, Germany;
     oleksandr.selyshchev@physik.tu-chemnitz.de (O.S.); o.raievska@fz-juelich.de (O.R.);
     zahn@physik.tu-chemnitz.de (D.R.T.Z.)
[4]  Center for Materials, Architectures and Integration of Nanomembranes (MAIN), Chemnitz University of
     Technology, 09107 Chemnitz, Germany
[5]  Forschungszentrum Jülich GmbH, Helmholtz-Institut Erlangen Nürnberg für Erneuerbare
     Energien (HI ERN), Immerwahrstr. 2, 91058 Erlangen, Germany; alstroyuk@ukr.net
*   Correspondence: dzhagan@isp.kiev.ua

**Abstract:** Thin films of colloidal CZTS nanocrystals (NCs) synthesized using a "green" approach in water with a variation of the copper-to-tin ratio are investigated by Raman scattering, mid-infrared (molecular vibrations) and near-infrared (free carrier) absorption, X-ray photoemission spectroscopy (XPS), electrical conductivity, and conductive atomic force microscopy (cAFM). We determined the effect of the actual Cu content on the phonon spectra, electrical conductivity, and spectral parameters of the plasmon band. An increase in the electrical conductivity of the NC films upon annealing at 220 °C is explained by three factors: formation of a $Cu_xS$ nanophase at the CZTS NC surface, partial removal of ligands, and improved structural perfection. The presence of the $Cu_xS$ phase is concluded to be the determinant factor for the CZTS NC film conductivity. $Cu_xS$ can be reliably detected based on the analysis of the modified Auger parameter of copper, derived from XPS data and corroborated by Raman spectroscopy data. Partial removal of the ligand is concluded from the agreement of the core-level XPS and vibrational IR spectra. The degree of lattice perfection can be conveniently assessed from the Raman data as well. Further important information derived from a combination of photoelectron and optical data is the work function, ionization potential, and electron affinity of the NC films.

**Keywords:** kesterite; CZTS; Cu vacancy; $Cu_xS$; plasmon; non-stoichiometry; secondary phases; phonons

## 1. Introduction

$Cu_2ZnSn(S,Se)_4$ (CZTSSe) has emerged as a promising candidate for next-generation photovoltaic (PV) technologies [1,2]. Unlike the present leader in the field of chalcogenide photovoltaics, $Cu(In,Ga)Se_2$ (CIGS), CZTSSe compounds consist of earth-abundant elements, which can allow them to be more competitive in the market. Owing to its direct bandgap, CZTSSe requires a hundred-times thinner absorber layer compared to indirect-bandgap Silicon, currently prevailing in photovoltaics, thus further reducing the manufacturing costs of perspective PV devices.

Despite the rapid initial progress in CZTSSe PV, a limit of 12–13% device efficiency was achieved with negligible further improvement over the past decade due to remaining structural defects and secondary phases, which could not be controlled so far [1,3,4]. The content of copper was found to be one of the determinant factors for the electrical properties

of CZTSSe layers [5–8]. In a single-crystal $Cu_2ZnSnS_4$ (CZTS), even tiny variations of copper content around the stoichiometric value, from 1.8 to 2.1, result in an order-of-magnitude change of the conductivity [9]. However, different Cu content values were reported to be optimal for the CZTSSe absorber layer in a PV device structure, implying that a solution could be a CZTSSe film with a Cu content varying with its thickness [5]. One of the promising routes to obtain a thin absorber layer on a deliberate substrate can be the synthesis of nanocrystals (NCs) of desired compositions in colloidal solutions [10,11], with subsequent deposition of a targeted sequence of the NC layers on the substrate by printing, spin-, or spray-coating [12].

Electrical properties of bulk single-crystal $Cu_2ZnSnS_4$ (CZTS) indicate p-type conductivity due to copper vacancies, with a specific resistivity of 16–100 $\Omega \cdot cm$ depending on stoichiometry [13,14]. The electrical behavior of microcrystalline CZTS thin films is different from that of the bulk material and significantly depends on the preparation method and the use of annealing. In general, annealing of CZTS thin films was shown to decrease resistivity, which can be explained by an increase in crystallinity, phase changing, increasing film homogeneity due to sintering of separate crystallites, or simultaneous appearance of these factors. The transition temperature from ordered to disordered CZTS kesterite takes place at T = 260 °C in an inert atmosphere [15]. On the other hand, the SnS phase begins to evaporate in a vacuum above 350 °C, and finally, from 550 °C, CZTS is not stable [16]. For CZTS thin films prepared with different methods, such as sol-gel, spray pyrolysis, magnetron sputtering, and "ink" solution deposition, a decrease in resistance of down to 0.2 $\Omega \cdot cm$ is observed when using different annealing parameters in each specific case [17–20]. Electrical properties of CZTS films prepared with colloidal nanocrystals (NCs) of sizes as small as several nanometers are less studied. Of particular interest is the effect of the annealing process on the ligand (in our case, small-sized TGA molecules), which is novel and usually not considered as a factor influencing the electrical properties of microcrystalline films. Soft annealing of NC films at the temperature of the boiling point of the TGA ligand is likely to (at least partially) remove the ligands and increase the integrity of the film, as well as increase the crystallinity of the NCs without disrupting them. Investigating the latter is a complex task, which, in particular, can be solved by vibrational Raman spectroscopy.

Raman spectroscopy is an established characterization technique for CZTSSe, CZTS, CZTSe, and many related compounds, including colloidal NCs [4,21–29]. It can detect secondary (impurity) phases [24,30–32] and point out defects [4,22,33], is not demanding regarding the amount of material, requires no special sample preparation for the measurement, unlike X-ray diffraction (XRD) and transmission electron microscopy (TEM), and can probe NCs even in as-synthesized solutions. It should be noted that despite impressive progress in comprehending the relationship between the phonon Raman spectra and the underlying CZTSSe structure in the last decade, making this technique as indispensable as XRD, there are still numerous puzzling observations that still need an unambiguous explanation. In particular, this concerns the intensity ratio of the kesterite modes as a measure of a particular defect density, as well as the quantitative relationship between non-stoichiometry and phonon spectra. On the one hand, there is a series of reports from several groups demonstrating a pronounced correlation between the intensity ratio of several Raman bands and the concentration of the $[V_{Cu} + Zn_{Cu}]$ defect clusters [4,21,33–36]. On the other hand, in a majority of the works that studied Raman spectra as a function of CZTSSe off-stoichiometry, no such distinct effect on the intensity ratio was observed [6,7,33].

Previously, we reported the aqueous synthesis of CZTS NCs stabilized by a small ligand, namely thioglycolic acid (TGA) [37], optimized in terms of their colloidal stability, size homogeneity, and degree of structural perfection concluded from XRD and Raman spectra. Even though X-ray photoelectron spectroscopy (XPS) revealed a remarkable deficit of Cu and Sn, XRD and Raman proved the excellent phase purity of those NCs, being as small as 3–4 nm. They revealed characteristic features of kesterite CZTS in XRD and no indications of secondary phases, as proved by Raman (except tiny contributions of ZnS and ZnO observable only at resonant UV excitation [30]).

Here, we report a study of CZTS NCs synthesized using the same mild approach with a variation of the copper composition. The series of NC samples with different Cu-content is characterized by Raman scattering, mid-infrared (molecular vibrations) and near-infrared (free carrier) absorption, XPS, electrical conductivity, and conductive atomic force microscopy (cAFM).

## 2. Experimental Section

CZTS colloids were produced via the reaction between sodium sulfide and a mixture of thioglycolate complexes of copper, tin, and zinc in aqueous alkaline solutions in the presence of ambient air at 22–24 °C and normal pressure. In a typical synthesis (referred to as sample "Cu$_2$", corresponding to the nominally stoichiometric Cu$_2$ZnSnS$_4$), 0.3 mL aqueous 1.0 M solution of Cu(NO$_3$)$_2$, 0.3 mL aqueous 0.5 M solution of freshly prepared SnCl$_2$ (containing 4.0 M NaOH), and 0.15 mL aqueous 1.0 M solution of Zn(NO$_3$)$_2$ were added consecutively to 6.0 mL deionized (DI) water with intense stirring followed by 3.0 mL of aqueous 1.0 M solution of TGA and 0.1 mL aqueous 10.0 M solution of NaOH. Finally, 0.3 mL aqueous 1.0 M solution of Na$_2$S was added lump-wise to this mixture with intense stirring. The final concentrations of the reactants were 0.03 M Cu(NO$_3$)$_2$, 0.015 M SnCl$_2$, 0.015 M Zn(NO$_3$)$_2$, 0.22 M NaOH, 0.3 M TGA, and 0.03 M Na$_2$S. In order to increase the crystallinity of the NCs, the as-prepared colloidal solution was then placed in a cylindrical glass vial with a diameter of 10 mm and heated in air conditions without stirring in a boiling water bath at 96–98 °C for 10 min. After the heat treatment, the CZTS colloid was subjected to purification from the residual salts by precipitation/redispersion. For this, 1 mL of 2-propanol was added to 10 mL of the CZTS colloid, and the mixture was subjected to centrifugation at 6000 rpm for 5 min. The wet precipitate was separated from the supernatant and redissolved in DI water (the total volume of the redispersed solution reached 2 mL) in an ultrasonic bath. As shown in [37], copper(II) and tin(II) change their valent state during the formation of NCs to Cu(I) and Sn(IV), respectively.

The Cu$_x$ZnSnS$_4$ NC samples with lower (x = 0.25) and higher (x = 4) copper content were obtained by a similar procedure with a load of 0.038 and 0.6 mL aqueous 1.0 M solution of Cu(NO$_3$)$_2$, respectively. The obtained light-brown and dark-brown colloids were subjected to purification as described above.

For the spectral Raman and mid-infrared studies, the freshly synthesized and purified samples were drop-casted on cleaned double side polished Si(100) substrates and dried in a desiccator under dynamic vacuum (~10 mbar). Fabrication of NC thin films for electrical measurements was performed by spin-coating the colloidal solutions on cleaned soda-lime 1 cm × 1 cm glass substrates. Three deposition cycles, each at 2000 rpm for 20 s, were applied. After each cycle, the samples were dried in ambient air at (120 ± 5) °C for 5 min to remove the moisture. Annealed NC films were obtained from the pristine films by annealing at (220 ± 5) °C for 60 min in the air if nothing else is specified. The temperature of 220 °C, which is the boiling point of thioglycolic acid at atmospheric pressure [38] (120 °C at 20 mmHg [39]) according to our initial hypothesis, should be sufficient to partially remove the ligands without significant NCs degradation. For electrical measurements, Au contacts were deposited using a shadow mask providing a pattern of electrodes and contact pads, as shown in Figure S1. Conductive AFM, as well as near-infrared, visible, and photoemission spectroscopic studies, were performed on the same samples as the electrical measurements.

Raman spectra were excited using a 514.7 nm diode-pumped solid-state (DPSS) laser (Cobolt, HÜBNER Photonics GmbH, Kassel, Germany) or 325 nm He–Cd laser line and registered at a spectral resolution of about 2 cm$^{-1}$ for visible and 5 cm$^{-1}$ for UV excitation using a LabRam HR800 micro-Raman system equipped with a liquid nitrogen-cooled CCD detector. The incident laser power under the microscope objective (50× for visible and 40× for UV light) was varied in the range of 0.1–0.001 mW.

IR spectra were collected in a transmittance geometry using a vacuum VERTEX 80v FTIR spectrometer (Bruker) equipped with an RT-DLaTGS detector and KBr beamsplitter for the middle infrared (MIR) range (400–4000 cm$^{-1}$), an InGaAs detector and a CaF$_2$

beamsplitter for the near-infrared (NIR) range (4000–11,000 cm$^{-1}$), and a Si detector and a CaF$_2$ beamsplitter for a NIR-visible range (11,000–21,000 cm$^{-1}$). The samples for MIR and NIR measurements were deposited on double side polished Si substrates. NIR-vis measurements were performed on the films deposited on glass substrates.

XPS measurements were performed with an ESCALAB 250Xi X-ray Photoelectron Spectrometer (Thermo Scientific, Waltham, MA, USA) equipped with a monochromatic Al K$\alpha$ (h$\nu$ = 1486.7 eV) X-ray source. A pass energy of 200 eV was used for survey spectra, 40 eV for Auger spectra, and 20 eV for high-resolution core-level spectra (providing a spectral resolution of 0.5 eV). Secondary electron cut-off (SECO) spectra were taken under a bias of $-10.0$ V, then the scale was corrected by 10 eV. Spectra deconvolution and quantification were performed using the Avantage Data System (Thermo Scientific). The valence band spectra were processed using the UNIFIT$^{TM}$ (2018) software. The valence band maximum (VBM) was obtained as a convolution of a linear and a Gaussian function with an FWHM of (0.55 $\pm$ 0.05) eV addressing the instrumental resolution. The energy scale was calibrated with an accuracy of $\pm 0.05$ eV using the binding energies (BE) of Au4f$_{7/2}$ at 83.95 eV, Ag3d$_{5/2}$ at 368.20 eV, Cu2p$_{3/2}$ at 932.60 eV, and the Fermi edge at 0.00 eV, measured on in situ cleaned metal surfaces. The spectra of NC thin films with sufficient conductivity (Cu$_2$ and Cu$_4$ both before and after annealing) were measured without an additional charge compensation setup. The low conductive NC thin films (Cu$_{0.25}$ before and after annealing) were measured with a built-in charge compensation system. If necessary, the spectra were corrected to the C1s sp$^3$ peak at 284.8 eV as the common internal standard for BE calibration [40].

Surface topology and local conductivity were investigated using a Ntegra AFM scanning probe microscope (NT-MDT). Conductive atomic force (cAFM) measurements were carried out using a tip coated with nitrogen-doped diamond (DCP11) under 1.5 V bias voltage applied between the tip and the electrode on the CZTS surface.

The resistivity of the thin films at different temperatures was investigated using a digital DLTS spectrometer FT-1030 by applying 100 mV bias voltage between two Au electrodes evaporated thermally on the CZTS NC film surfaces. A four-point probe was used to determine resistivity and I–V curves at room temperature. The I–V curves were found to be linear in the range from $-1$ V to $+1$ V thus revealing Ohmic behavior.

## 3. Results and Discussion

### 3.1. X-ray Photoemission Spectroscopy

The XPS measurements of the thin films of CZTS NCs with different nominal Cu content confirm that the targeted trend in the NCs composition was achieved (Figures 1 and S2, Table S1). It can be seen from the XPS data normalized to sulfur (Figure 1), that in the initial NC films the amount of incorporated copper matches well with the nominal load, especially for the stoichiometric value, x$_{Cu}$ = (1.9 $\pm$ 0.2) for the sample Cu$_2$. For the samples with lower (Cu$_{0.25}$) and higher (Cu$_4$) nominal loads, the actual Cu content is higher (x$_{Cu}$ = 0.8 $\pm$ 0.1) and lower (x$_{Cu}$ = 3.2 $\pm$ 0.3), respectively, than the nominal amounts. This apparent deviation can be caused by the non-stoichiometry of the CZTS phase, well known from numerous previous studies [3,28], and/or by inclusions of ternary and binary secondary phases. These issues are discussed in more detail in the Raman part of this work. It should be noted that all the CZTS NC films studied here were Zn-rich and Sn-deficient (Figure 1), including the nominally stoichiometric sample, Cu$_2$. Annealing at 220 °C affects the composition of the film surface for all samples resulting in a decrease in the Cu content of Cu$_{0.25}$, Cu$_2$, and Cu$_4$ to x$_{Cu}$ = (0.4 $\pm$ 0.1), (1.2 $\pm$ 0.1), and (1.8 $\pm$ 0.2), respectively. The Sn content increases for all samples, while the trend is not monotonic for Zn. The Zn content decreases for the films Cu$_{0.25}$ and Cu$_2$, while for the sample Cu$_4$, the amount of Zn increases more than two times. Since the sampling depth for photoelectrons emitted under Al K$_\alpha$ excitation is less than 10 nm [40], such behavior indicates dramatic changes in the composition of the surface layer due to element diffusion or chemical transformation.

Particularly, high Zn content can be due to the ZnS secondary phase, as revealed by UV Raman spectra.

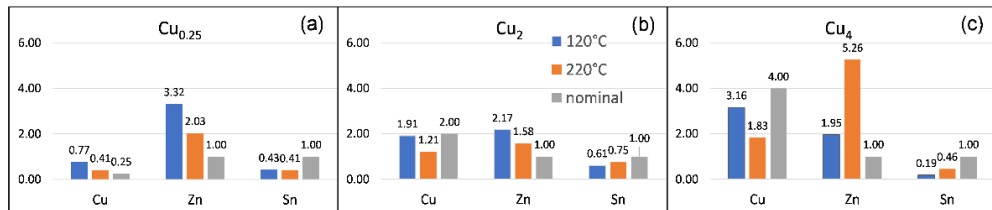

**Figure 1.** Actual (XPS-derived) and nominal (loaded during synthesis) ratios of metallic elements normalized with respect to the sulfur content ($S_4$) of the CZTS NC films with different Cu content for the initial (120 °C) and annealed (220 °C) samples: (**a**) $Cu_{0.25}$; (**b**) $Cu_2$; (**c**) $Cu_4$. The compositions in atomic % can be found in the Supplementary Materials (Table S1).

It was established previously that the core-level XPS spectra of Cu do not allow the CZTS and $Cu_xS$ phases to be distinguished unambiguously [30,41]. Much more informative is the analysis of the so-called modified Auger parameter $\alpha'_{Cu}$, calculated as the sum of the binding energy (BE) of the $Cu2p_{3/2}$ core level and the kinetic energy (KE) of the Cu LMM ($^1G_4$) Auger feature [30]. Since these two scales have opposite signs (BE = hν − KE, where hν is the energy of X-ray photons), the modified Auger parameter is independent of the charging effect. It can also be used as a characteristic fingerprint of a certain compound, for instance, allowing to distinguish $Cu_2ZnSnS_4$ from $Cu_2S$ and CuS [42,43]. In our case, the initial CZTS NC films with a nominal composition of $Cu_{0.25}$ show an $\alpha'_{Cu}$ typical for CZTS NCs (Table 1), while the $Cu_4$ film indicates the presence of $Cu_2S$. The film $Cu_2$ exhibits an intermediate $\alpha'_{Cu}$ value, indicating the presence of a mixture of CZTS and $Cu_2S$ phases. The $\alpha'_{Cu}$ value obtained for $Cu_2$, 1849.5 ± 0.2 eV, is higher than the one previously reported by us for similarly synthesized aqueous CZTS NCs ($\alpha'_{Cu}$ = 1848.8 ± 0.2 eV) [30]. The reason for the shift is the $Cu_2S$ secondary phase, formed upon the heating of the film at 120 °C, initially applied to the films in this work to remove the moisture before electrical characterization. Indeed, a reference CZTS NC sample without any thermal treatment showed the same $\alpha'_{Cu}$ value as in the previous work [30]. Taking into account the surface sensitivity of the XPS technique and the fact that Raman spectra were taken with visible excitation (vis-Raman) on the same films reveal only CZTS kesterite features (discussed below), we can conclude that for CZTS samples with a high Cu content, even mild heat treatment at 120 °C partially (for $Cu_2$) or fully (for $Cu_4$) converts the surface layer of CZTS to $Cu_2S$. At a temperature of 220 °C, the conversion rate increases. As a consequence, the annealed NC films $Cu_{0.25}$ and $Cu_2$ show a mixture of $Cu_2S$ and CZTS, while the sample $Cu_4$ indicates the formation of a sulfur-rich phase of copper sulfide, i.e., CuS (Table 1). It should be observed that a comparison of the Raman and XPS data indicates that the $Cu_2S$ and CuS secondary phases are predominantly formed in the surface layers and not in the entire volumes of the films. Moreover, Raman spectroscopy is known to be sensitive to CuS but not to $Cu_2S$ [44,45].

The sulfur chemical states are also affected by the annealing of the CZTS NC films with different copper contents. Figure 2 shows the S2p core level spectra of the $Cu_4$ films before and after annealing at 220 °C. In the initial NC film (i.e., dried at 120 °C), three major chemical components of sulfur are observed, previously assigned to the NC inorganic sulfide in the core ($S^{2-}$), boundary surface sulfur shared between inorganic sulfide and thioglycolate ligand ($S^{1-}$), and free, non-bonded to the NC surface, thiol sulfur ($S^0$) [30,39]. A fourth, minor component with an $S2p_{3/2}/S2p_{1/2}$ doublet at 167/168 eV assigned to sulfite ($S^{4+}$) is due to partial oxidation of the NC surface or thiolate ligand. Annealing at 220 °C leads to a drastic reduction of the contribution of the free-ligand doublet at 164/165 eV ($S^0$). Moreover, the component related to ligands (chemically) bound to the NC surface, at 165/166 eV ($S^{1-}$), also drops in intensity by a factor of two. The latter results indicate that the annealing of the films leads not only to the removal of most unbound

ligand molecules but also to a partial loss even of ligands bound to the NC surface. Finally, the sulfite sulfur ($S^{4+}$) is further oxidized to sulfate ($S^{6+}$), as can be concluded from the shift of the fourth doublet to higher binding energies at 169/170 eV.

**Table 1.** Copper sulfide secondary phase identification in the initial and annealed CZTS NC films, based on analysis of the modified Auger parameter $\alpha'_{Cu}$. The corresponding $Cu2p_{3/2}$ and Cu LMM Auger spectra can be found in the Supplementary Materials (Figure S3).

| | $Cu2p_{3/2}$/ $\pm0.1$ eV | Cu LMM ($^1G_4$)/$\pm0.1$ eV | $\alpha'_{Cu} = Cu2p_{3/2} + Cu_{LMM}$ ($^1G_4$)/$\pm0.2$ eV | Assignment | Literature Data, $\alpha'_{Cu}$/eV |
|---|---|---|---|---|---|
| $Cu_{0.25}$ (initial, 120 °C) | 931.7 | 917.0 | 1848.7 | CZTS | 1848.6–1849.2, CZTS [30,42] |
| $Cu_2$ (initial, 120 °C) | 932.5 | 917.0 | 1849.5 | $Cu_2S$ + CZTS | - |
| $Cu_4$ (initial, 120 °C) | 932.4 | 917.5 | 1849.9 | $Cu_2S$ | 1849.8–1849.9, $Cu_2S$ [40,42,43] |
| $Cu_{0.25}$ (annealed, 220 °C) | 932.4 | 917.1 | 1849.5 | $Cu_2S$ + CZTS | - |
| $Cu_2$ (annealed, 220 °C) | 932.2 | 917.5 | 1849.7 | $Cu_2S$ + CZTS | - |
| $Cu_4$ (annealed, 220 °C) | 932.2 | 917.9 | 1850.1 | CuS | 1850.0–1850.4, CuS [40,42,43] |

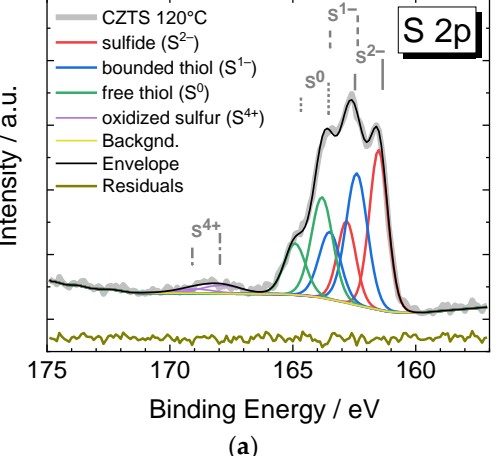 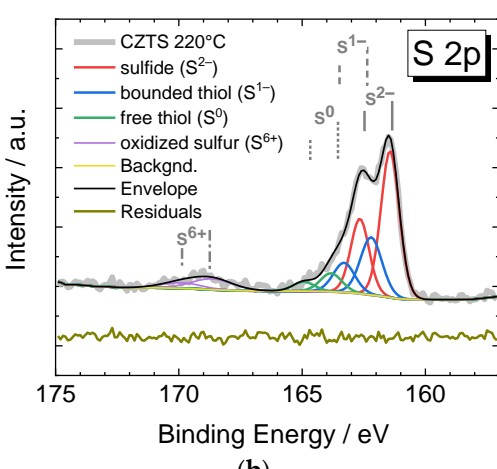

(**a**)            (**b**)

**Figure 2.** The S2p XPS spectra of CZTS NC $Cu_4$ films in the initial state (**a**) and after annealing in air (**b**).

Besides the chemical composition of the samples, photoelectron spectroscopy allows several other crucial parameters to be determined. The ionization potential (IP) of CZTS NC films, which is one of the key parameters for device applications, was determined from the positions of the secondary electron cut-off (SECO) and the valence band maximum (VBM) of the XPS spectra (Figure 3). The secondary electron cut-off also allows the value of the work function (WF) of the NC film surface to be determined (Figure 3a). The SECO spectra of $Cu_{0.25}$ samples provided unreasonable values, most likely due to using a charge compensation during the measurements, which applies a flux of electrons with low kinetic energy compared to the energy of the secondary electrons emitted from the sample. The observed proximity of the Fermi level to the VBM indicates that the films are p-type semiconductors in agreement with previous results [46] and the presence of copper sulfide on the surface, which is known to be a strong p-type semiconductor [46,47]. Because of the opposite signs of the scales used to represent the secondary and valence photoelectrons (Figure 3a,b), the ionization potential is independent of the charging effect, similar to a modified Auger parameter. Finally, the electron affinities (EA) are estimated by adding the bandgap values, determined from the optical spectra to the Ips. The resulting energy level diagrams vs. the vacuum level are shown in Figure 3c. The obtained EA values are relatively high, therefore the NC surface can be readily oxidized by air oxygen. In our

study, partial sulfur oxidation at the film surfaces ($S^{4+}$ and $S^{6+}$ chemical states in the S2p spectra) was detected. We assume that the oxide layer formed passivates the surface and prevents deeper oxidation. Nevertheless, our observation of the high EA for the $Cu_2S/CuS$ surface of CZTS NC films is not surprising because similarly high EA values for pure copper sulfides were reported previously [42,48].

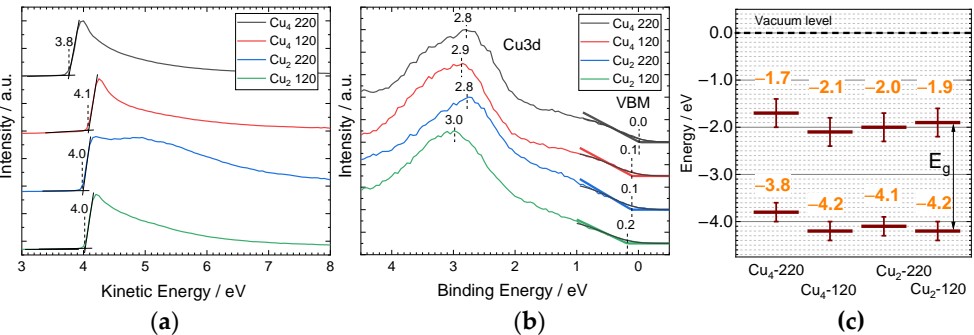

**Figure 3.** The secondary electron cut-off (SECO) (**a**) and valence band (VB) (**b**) XPS spectra of the initial and annealed CZTS NC films with the nominal contents $Cu_2$ and $Cu_4$. (**c**) Energy level diagrams derived from the XPS and optical spectroscopy data.

### 3.2. Raman Spectroscopy

Previously, Raman spectroscopy was shown to be one of the most informative methods regarding the structural and elemental composition of CZTS [21–24]. Figure 4 shows Raman spectra of the samples at $\lambda_{exc} = 514.7$ nm, which is resonant for the excitation of the phonons in CZTS, as well as for several secondary phases including CuS [23,24].

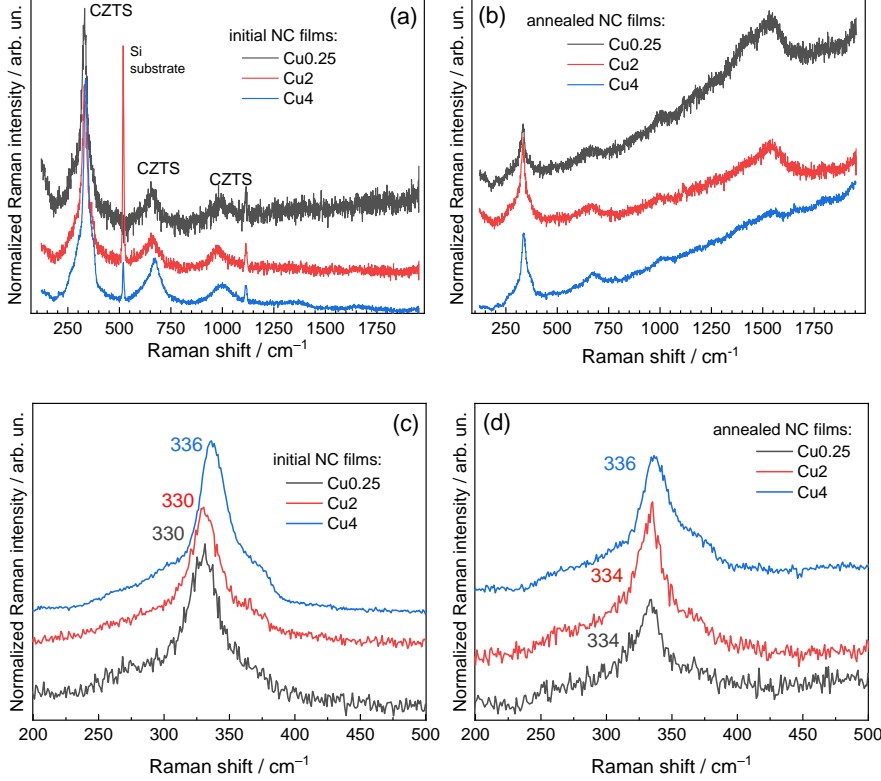

**Figure 4.** Raman spectra ($\lambda_{exc} = 514.7$ nm, 0.15 mW) of the initial (**a,c**) and annealed (**b,d**) CZTS NC films with different nominal Cu content. The spectra in a broad spectral range are shown in (**a,b**), while (**c,d**) shows in more detail the range of the first-order CZTS phonons. The sharp peak at 1100 cm$^{-1}$ is an artifact (a room light). In (**b**), the background, increasing toward larger wavenumbers, is due to PL of the carbonized ligand.

All peaks in the spectra in Figure 4a can be assigned to kesterite CZTS. In particular, the characteristic first-order peak around 330 cm$^{-1}$ of A$_1$ symmetry (corresponding to S-S vibration with the cations at rest) and the higher-order phonon scattering peaks around 660 and 990 cm$^{-1}$ agree well with the spectrum of crystalline bulk Cu$_2$ZnSnS$_4$ of the kesterite modification [23]. The Sn deficiency in our NC samples (Figure 1) is unlikely to be the reason for the A$_1$ peak broadening because a very sharp Raman peak at 337 cm$^{-1}$ was observed for both Sn-poor, Cu-rich, or Zn-rich CZTS [41]. The FWHM (~30 cm$^{-1}$) of the main peak is comparable to that of the NCs synthesized in high-boiling-point solvents [49–54], and the higher-order phonon features (at 658 and 995 cm$^{-1}$) indicate a good crystallinity of the CZTS NCs of small size, 3–4 nm [37], synthesized here under relatively mild conditions in the water.

From the frequency position of the main Raman mode of 336 cm$^{-1}$, we can assume that the NCs synthesized with x$_{Cu}$ = 3.2 (nominally Cu$_4$) possess the typical kesterite structure. The 330 cm$^{-1}$ peak frequency of nominally stoichiometric Cu$_2$ and Cu-poor Cu$_{0.25}$ samples indicates the formation of a so-called cation-disordered kesterite structure [55–59]. The same peak position for the initial NC samples with x$_{Cu}$ = 0.8 (Cu$_{0.25}$) and 1.9 (Cu$_2$) (Figure 4c) may indicate the fact that this Raman peak position is indeed determined by the type of the lattice structure/order, "ordered kesterite" or "disordered kesterite", and the variation of the (cation) composition within the range of stability of the particular phase has a minor effect. This behavior can be explained by the fact that the mode corresponding to the A$_1$-symmetry involves mainly (covalent) S-S vibration, and the cation environment may indeed have only a minor effect on its frequency.

The results of annealing further confirm the assumption above. Annealing the sample with the highest copper content, x$_{Cu}$ = 1.8 (nominally Cu$_4$, annealed), does not affect the peak position, 336 cm$^{-1}$, and the peak width (Figure 4d). Annealing the samples with x$_{Cu}$ = 0.4 (Cu$_{0.25}$) and 1.2 (Cu$_2$) shifts the Raman peak from 330 up to 334 cm$^{-1}$, also without a noticeable narrowing of the peak. This shift can be understood as bringing more order in the cationic sublattice, i.e., transforming the structure of the NCs from "disordered kesterite" to "ordered kesterite". Note that XPS reveals a decrease in the copper content upon annealing. However, this does not contradict putting more cations into the right lattice sites. Moreover, XPS mainly probes the surface (<10 nm), while Raman probes at least 100 nm film thickness.

From observing Raman bands at 1300–1600 cm$^{-1}$ (Figure 4b), which are characteristic of amorphous carbon [60], one can conclude the (partial) decomposition of the ligands upon annealing. However, the ligand carbonization is only partial, because the FTIR measurements (discussed in the next section), which are more quantitative than the Raman ones, show the intensity of ν(COO$^-$) modes, proportional to the ligand content, decreasing but not fully disappearing upon annealing. The background slope appearing in the Raman spectra upon annealing can be related to the photoluminescence (PL) of the amorphous carbon as well [61].

The Raman spectra presented in Figure 5 (for the Cu$_2$ sample as an example) reveal no effect of oxygen and water contained in the air during the annealing of the CZTS NC films on the phase composition of the NC films. Surprisingly, no remarkable difference between oxygen-containing and inert annealing environments, both at λ$_{exc}$ = 514.7 nm and 325 nm, were observed. The distinct difference between spectra taken with "green" and "UV" excitations has two reasons: resonance effects and much shallower probing depth of the 325 nm light than that for 514.7 nm.

The spectra at "green" excitation reveal amorphous carbon Raman bands and PL background after annealing in both air and N$_2$ conditions. The PL can be related to the forming of small amorphous clusters (carbon dots) [62]. No new peaks arise, which could indicate the formation of secondary phases of Cu$_x$S or Cu$_x$SnS$_y$. On the other hand, weak peaks related to Cu$_x$S, ZnS, and ZnO dominate the spectra at UV excitation. Observing ZnO and ZnS is not surprising because these wide-bandgap compounds can be resonantly probed at UV excitation, as already proved in previous works on CZTS and other multinary

Zn-containing NCs [23,27]. The $Cu_xS$, however, is usually much better detected at green excitation [23,24,44]. In order to better understand this uncommon observation, we studied the same three samples at different laser powers at $\lambda_{exc}$ = 325 nm (Figure 6).

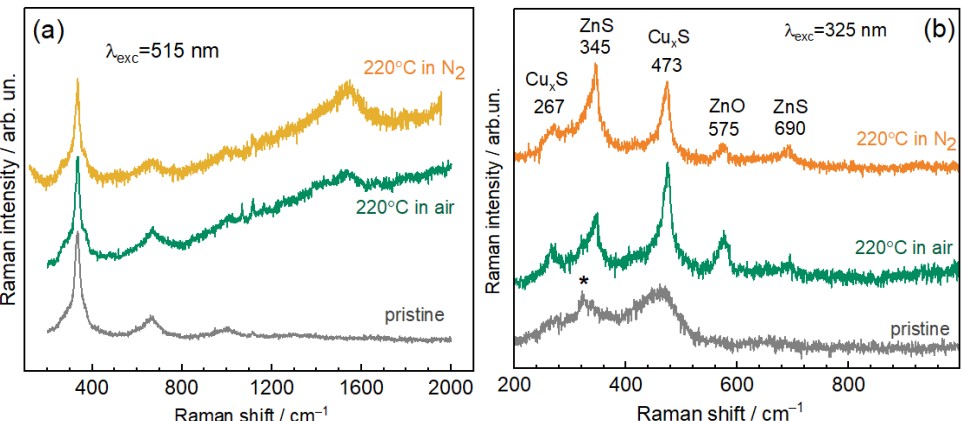

**Figure 5.** Raman spectra of the CZTS NC films with the nominal content $Cu_2$ measured with $\lambda_{exc}$ = 514.7 nm (**a**) and $\lambda_{exc}$ = 325 nm (**b**) on the pristine film and films annealed at 220 °C in air and in $N_2$. The feature marked by an asterisk in (**b**) is an artifact due to Raman scattering in the UV objective.

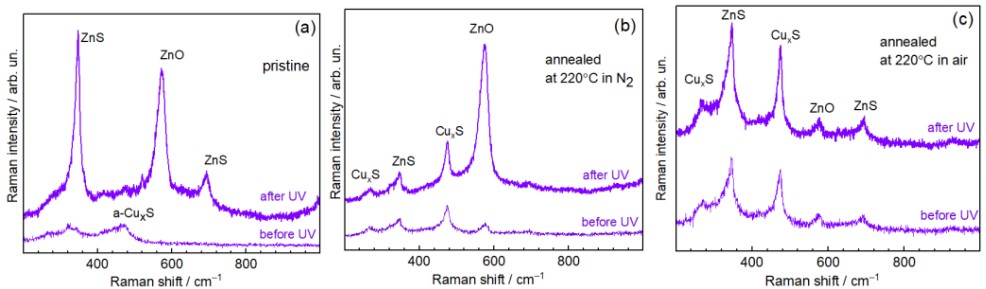

**Figure 6.** The effect of UV illumination on the Raman spectra of the CZTS NCs with the nominal content $Cu_2$ for the initial film (**a**) and films annealed in air (**b**) and in $N_2$ (**c**). The spectra measured with 0.1 mW of the $\lambda_{exc}$ = 325 nm before (bottom curves) and after (upper curves) illumination at 0.5 mW for 5 min.

We find that the formation of secondary phases in the initial and $N_2$-annealed NCs can be stimulated by increasing the excitation power of the UV laser. The NC film annealed in air, on the contrary, exhibits better resistance to the photo-induced formation of secondary phases. However, the content of these phases in the initial sample before UV is comparable to the sample annealed in $N_2$ (as can be inferred from the comparable intensity of Raman peaks and signal-to-noise ratio in the bottom curves in Figure 6b,c). We already reported the excitation dependence of secondary $Cu_xS$ phase formation in CZTS in our previous paper [30]. In particular, more $Cu_xS$ was generated with shorter $\lambda_{exc}$. Those results agree with the stronger photo-induced formation of the $Cu_xS$ phase at UV excitation compared to the "green" excitation.

### 3.3. Infrared Spectroscopy

Mid-infrared spectra allow us to register vibrational modes of ligands in the NCs samples, namely the thioglycolic acid and/or thioglycolate metal salt derivatives, thus allowing one to monitor the evolution of the organic part of the NC film with annealing (Figure 7). The spectral region of 700–1100 cm$^{-1}$, corresponding to C-C stretching and C-H deformational bonds, is known as a "fingerprint" region for many organic compounds [63]. However, in our case, the intense stretching vibrations of O-C=O fragments detected in the range of 1250–1750 cm$^{-1}$ are of particular interest, since they allow for distinguish-

ing between carboxylic (COOH) and carboxylate (COO⁻) groups and even determining the type of bonding between oxygen atoms and metal ions [63]. A protonated COOH group contributes to an FTIR spectrum with two individual C-O and C=O vibrations, experimentally observed for liquid TGA at 1295 and 1715 $cm^{-1}$, respectively (Figure 7d).

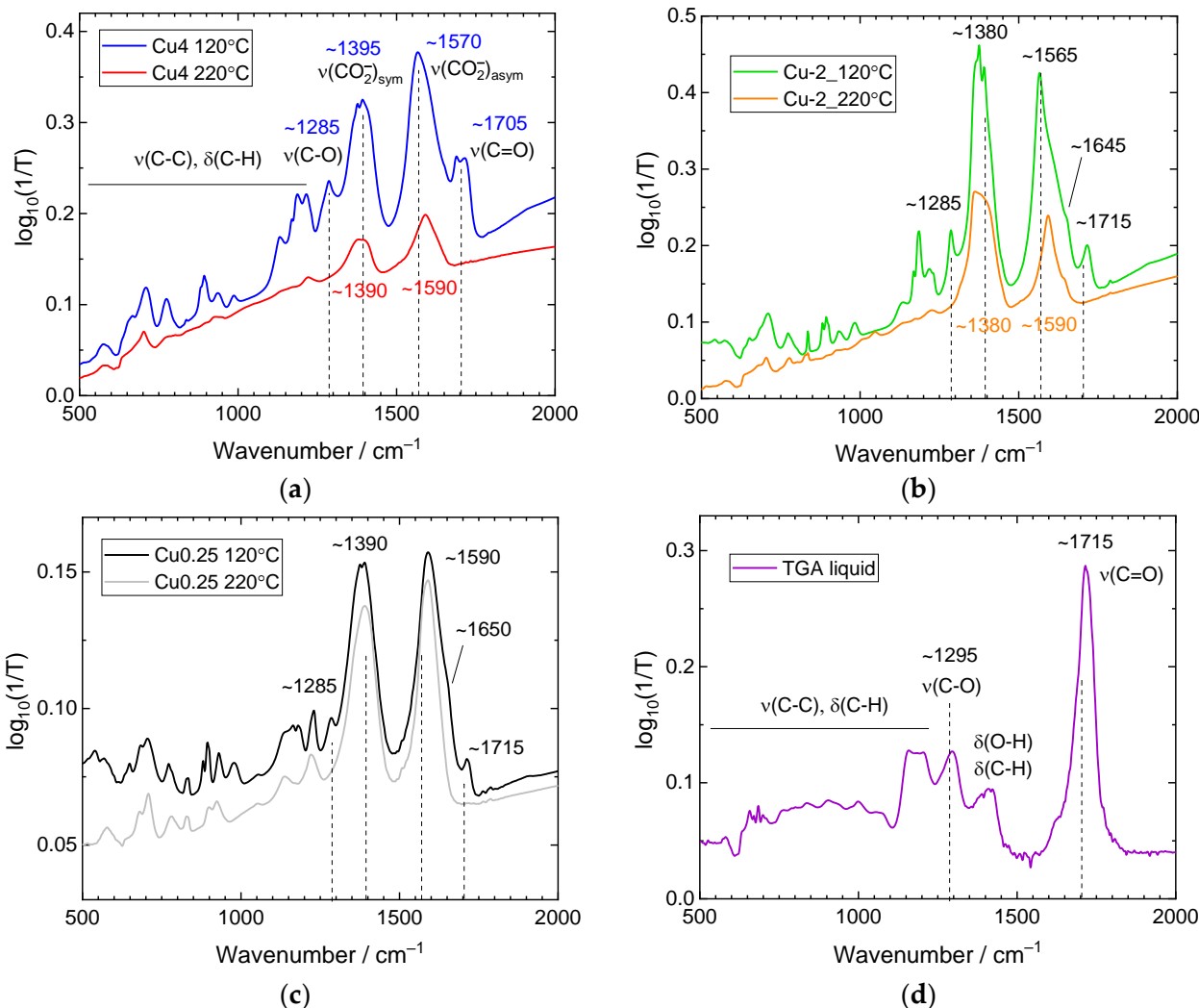

**Figure 7.** Mid-FTIR spectra of the initial and annealed CZTS NC films with the nominal contents $Cu_4$ (**a**), $Cu_2$ (**b**), and $Cu_{0.25}$ (**c**). The spectrum of the pure ligand (TGA) is shown for comparison (**d**).

Similar bands are detected in all CZTS samples before annealing. For a deprotonated COO⁻ group, the electron density between C-O and C=O is delocalized, making these bonds indistinguishable and contributing to the FTIR spectrum with symmetric (lower wavenumbers) and asymmetric (higher wavenumbers) vibrations of all three atoms [63]. The frequencies of these two vibrations and the splitting Δ between them partially depend on the heavy atom (metal ion) mass, but the type of bonding has an even greater influence on the metal ion, determined by the size, charge, and coordination abilities of the latter. Comparing our spectra with the literature data on Cu-, Zn-, and Sn-carboxylates [63,64], we can conclude that, before annealing, the $Cu_4$ and $Cu_2$ samples reveal a chelate bonding of thioglycolate to Zn and Cu (the $\nu(COO^-)_{sym}/\nu(COO^-)_{asym}$ doublet at ~1395/~1570 $cm^{-1}$, respectively). The $Cu_{0.25}$ sample demonstrates a bridging type of COO⁻ bonding to Cu and Zn (~1395/1590 $cm^{-1}$). The TGA derivative in the sample $Cu_2$ is additionally bonded to Sn (~1395/1645 $cm^{-1}$). After annealing at 220 °C, the bonds between COO⁻ and Cu/Zn are predominantly bridging, while other bonding types are suppressed or disappear. Another effect of annealing is the elimination of the protonated COOH groups, most likely due to

their chemical reaction with metal ions or degradation of the ligand. It should be noted that the XPS study clearly shows the bonding of the TGA ligands to the NC surface via the S-side ($S^{-1}$ chemical component), in agreement with the strong chemical affinity between sulfur and heavy metal atoms [65]. In turn, the FTIR results show that only part of the carboxylic groups of the TGA molecules are protonated and the rest of the molecules are deprotonated and coordinated to metal ions. The discrepancy can be due to a more complex structure of the ligand shell. For example, TGA is bonded to the NC surface via the thiol S, and the deprotonated $COO^-$ group binds to one of the ions ($Zn^{2+}$, $Cu^{2+}$, $Cu^+$, $Sn^{4+}$, or even $Na^+$) available in the colloidal solution during the NC synthesis.

The intensity of the infrared spectra in Figure 7 shows quantitative changes in the ligand content because they are taken on the same samples before and after annealing, covering the entire sample area and maintaining the same optical path. The reduced intensity of the FTIR peaks after annealing indicates fewer TGA derivatives in the samples, suggesting partial decomposition of the TGA ligand. It can be seen that for the $Cu_4$ sample, the ligand content is significantly reduced, the $Cu_{0.25}$ sample is negligibly affected, and the $Cu_2$ sample is an intermediate case. Therefore, we can conclude that the ligand decomposition (elimination) during annealing is proportional to the copper content in the CZTS NC films. It should be noted that the $Cu_4$ sample possesses the least amount of ligands after annealing and, at the same time, shows the most significant conductivity. Thus, one of the reasons for the increase in conductivity of the CZTS NC films after annealing can be an improvement in the number of contacts and the shorter distances between NCs and/or carbonization of the ligands as a result of ligand pyrolysis (decomposition).

### 3.4. Electrical Conductivity of the NC Films

Figure 8 shows the temperature dependence of resistivity of the CZTS NC thin films with various Cu contents. Several conductivity mechanisms are usually observed in the CZTS films [66]. At temperatures >100 K, the conductivity is determined mainly by the concentration of free holes released from defect states and the thermal emission of holes over the potential barriers. As the temperature decreases, the conductivity is dominated by the Mott variable range hopping and nearest-neighbor hopping mechanisms [66]. A high concentration of native defects may lead to Fermi level pinning and be responsible for the high conductivity of the films studied and their strong dependence on the Cu content.

The conductivity of the NC films increases by a few orders of magnitude after annealing at 220 °C. Two principal factors can be assumed to lead to higher conductivity. First, a larger number of lattice sites contribute free charge carriers, i.e., Cu vacancies in the case of CZTS, $Cu_xS$, or other non-stoichiometric Cu-chalcogenides [66,67]. Our XPS results support this option, showing a lower Cu content upon annealing (Figure 2). Another factor could be the ordering of the cationic lattice upon annealing, concluded above from the narrowing of the phonon Raman peak. A certain reduction of the free charge carrier concentration due to partial elimination of Cu vacancies upon annealing, revealed by NIR absorption discussed below, is apparently not detrimental for the resultant increased conductivity of the film due to the first factor. A reduced concentration of the recombination sites and improved interparticle transport due to the formation of highly conductive $Cu_xS$ shells of CZTS NCs are more significant benefits of the annealing for the net electrical conductivity. A possible better connection between neighboring NCs upon annealing could also be due to a decreased number of ligands between the NCs and their partial conversion into amorphous carbon (which is presumably more conductive than the pristine ligand molecule, TGA), as shown above by IR and Raman spectroscopies.

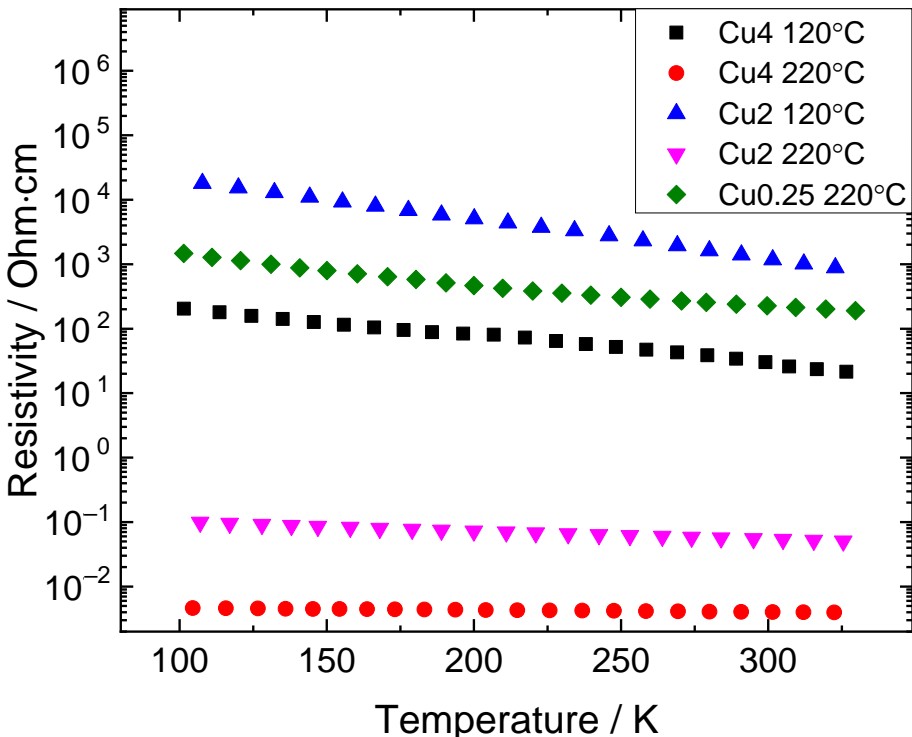

**Figure 8.** Temperature dependence of the resistivity of the initial (i.e., dried at 120 °C) and annealed at 220 °C in air CZTS NC thin films with various Cu contents.

The changes in charge transport properties upon annealing were detected at the nanoscale by cAFM measurements. The conductivity of the initial samples was not sufficient to be detected with a conductive tip. At the same time, the cAFM maps presented for the annealed samples show a non-uniform spatial distribution of the local conductivity. The sample with the highest Cu content, $x_{Cu}$ = 1.8 (Cu$_4$ after annealing), has a better conductivity of the grains than regions between neighboring crystalline areas and a good correlation of the NC position and current density. This probably originates from the better structural quality of CZTS NCs with a Cu content close to the stoichiometric one and the presence of the Cu$_x$S phase at the NC surface. For the sample with lower Cu content, the cAFM maps become more inhomogeneous, with the lowest conductivity of large grains and the highest conductivity of the surrounding areas. The macroscopic conductivity of the Cu$_2$ sample is lower than that of Cu$_4$ due to the presence of nonconductive "spots" (see Figure 9d), which is probably a consequence of the deficiency of Cu and non-uniform spatial distribution of the conductive secondary phase Cu$_x$S at the NC surface. The XPS data confirm the secondary phase formation on the surface during annealing. Obviously, its contribution to the conductivity is more significant in samples with a high content of Cu ($x_{Cu}$ = 1.8), while the deficiency of Cu in a sample with $x_{Cu}$ = 1.2 leads to its local high resistance in some areas of the surface and inferior macroscopic conductivity.

*3.5. Optical Absorption*

Based on earlier systematic studies of the relationship between the plasmon peak position and the concentration of Cu vacancies (V$_{Cu}$) in Cu$_x$S (or Cu$_x$Se) NCs [26,67–69], the redshift of the plasmon peak upon annealing of the CZTS NC films in this work (Figure 10a) indicates a decrease in the free carrier concentration possibly related to a decreasing concentration of Cu vacancies (V$_{Cu}$). Similarly to the localized surface plasmon resonance (LSPR) in metal NCs, the spectral position of the plasmon band of the p-type carriers in vacancy-doped NCs is determined by the carrier (hole) concentration: $\omega_{sp} = \frac{1}{2\pi}\sqrt{Ne^2/\varepsilon_0 m_e(\varepsilon_\infty + 2\varepsilon_m)}$ [67]. From this equation, the effective concentration of the free holes in the NCs can be evaluated: N~$10^{21}$ cm$^{-3}$, where N is charge carrier (hole)

density, *e* is the electron charge, $\varepsilon_0$ is the permittivity of vacuum, $m_e$ is the mass of electron, $\varepsilon_\infty$ is the high frequency dielectric constant of NC material, $\varepsilon_m$ is the dielectric constant of the NC environment.

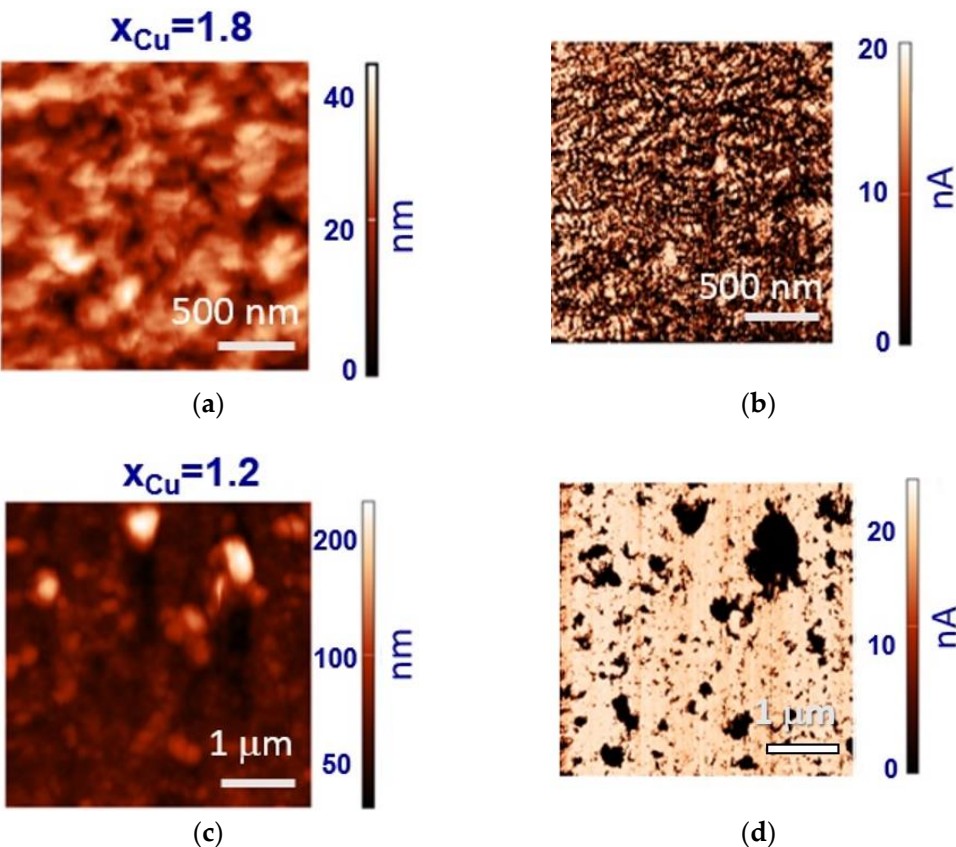

**Figure 9.** The topology (left) and cAFM maps (right) for samples with $x_{Cu} = 2$ (Cu₄) (**a,b**) and $x_{Cu} = 0.7$ (Cu₂) (**c,d**) after annealing. The applied bias is 1.5 V.

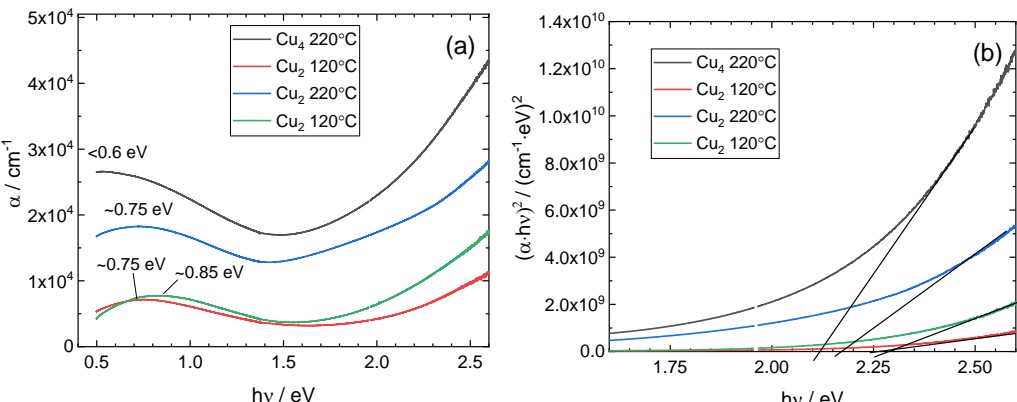

**Figure 10.** (**a**) Optical absorption in the near-infrared range exhibiting plasmon absorption by free charge carriers in the NCs. (**b**) Optical absorption in the visible, with the approximate position of the absorption edge indicated.

The size of the CZTS NCs used in this work was determined earlier to be 3–4 nm [37]. Based on this fact, the size of the $Cu_xS$ that form as a secondary phase is unlikely to be much larger after relatively soft annealing (220 °C). The plasmon peak position reported in the literature for 3–6 nm $Cu_xS$ is 0.5–0.6 eV [67].

The determination of the absorption edge of the NCs is not straightforward (Figure 10b). As is common for semiconductor NCs, the position of the fundamental bandgap and absorp-

tion edge is determined by the NCs size due to quantum confinement [70]. The bandgap and absorption edge energies do not differ much for highly crystalline and stoichiometric NCs, such as II–VI ones, which exhibit distinct absorption onset and peaks due to interband optical transitions even for small NC sizes (<5 nm [71,72] or even <2 nm [73,74]) and synthesized in water at low temperature (<100 °C) [72,74]. For non-stoichiometric and/or cation-disordered NCs of I–III–VI and I–II–IV–VI compounds rich in cation vacancies, the absorption onset is very smooth/smeared, making the determination of the bandgap very inaccurate or hardly possible [75]. The bandgap values reported in the literature for 3–6 nm $Cu_xS$ NCs are 1.5–1.7 eV [67]. From Figure 10a it may be concluded that the interband absorption starts in the range where the high-energy tail of the plasmon peak ends. With the optical spectra plotted in the Tauc coordinates for direct transition, one can approximate the absorption edge in the range of 2.15–2.25 eV (Figure 10b), which can correspond to the interband absorption of the CZTS NC phase [37].

The decrease in the concentration of Cu vacancies ($V_{Cu}$) upon annealing, concluded above from the plasmon peak behavior (Figure 10a), is in good agreement with the results from Raman spectroscopy discussed in the previous section. In particular, the narrowing of the phonon peak and its shift to the wavenumbers characteristic of ordered kesterite structure corroborate a decrease in the $V_{Cu}$ in the lattice.

## 4. Conclusions

Thin films of colloidal CZTS NCs synthesized using a "green" approach in water with a variation of copper content were investigated by a set of experimental techniques to establish the relationship between the copper content and their electronic, electrical, optical, and vibrational properties. Raman scattering reveals changes in the CZTS phase structure and the presence of secondary phases, particularly for $Cu_xS$. Mid-infrared absorption by molecular vibrations reveals a transformation in the ligand content, the near-infrared absorption shows free carrier (plasmon) band behavior, while absorption in the visible spectral range allows the optical absorption edge to be determined. From the XPS data, the elemental composition of the film surfaces, the presence of $Cu_xS$ phases, and the positions of energy levels confirming a strong p-type semiconductor character are determined. Macroscopic conductivity and conductive AFM mapping monitored the changes in the electrical conductivity of the NC films. From the analysis of these spectroscopic and microscopic results, the increase in the electrical conductivity of the NC films with an increase in Cu content and upon annealing (at 220 °C) is observed. This is explained by three factors: the formation of the conductive $Cu_xS$ nanophase, partial ligand removal, and improved structural perfection. The presence of a $Cu_xS$ phase at the NC surface was concluded based on the analysis of the modified Auger parameter of copper, derived from XPS data, and corroborated by Raman spectroscopy data. Partial ligand removal is concluded from the agreement of the core-level XPS and vibrational IR spectra. Further important information derived from a combination of photoelectron and optical data is the work function, ionization potential, and electron affinity of the NC films. Finally, an unambiguous dependence of the ligand removal efficiency on the actual Cu content is established.

**Supplementary Materials:** The following supporting information can be downloaded at: https://www.mdpi.com/article/10.3390/electronicmat3010013/s1, Figure S1: A representative photograph of the sample used for electrical measurements of the CZTS NC films in this work. NC film is deposited on a glass substrate by spin-coating, gold contacts deposited on the NC film by thermal evaporation, silver paste on top of the gold contact pads, and thermal paste on the bottom of the substrate; Figure S2: XPS-derived copper contents in CZTS films, normalized for sulfur content ($S_4$) for comparison with the stoichiometric kesterite $Cu_2ZnSnS_4$. The data correspond to the films heated at 120 °C (initial) and 220 °C (annealed). The dashed line is an eye guide for the ideal case of matching the measured amount of copper with its nominal load, Figure S3: Extended Cu2p (a), narrow Cu2p$_{3/2}$ (b), and Cu$_{LMM}$ Auger (c) high-resolution XPS spectra of pristine and annealed CZTS NC films synthesized with different nominal copper contents; Table S1: Elemental compositions

(in atomic %) derived from XPS on the CZTS NC samples with different nominal Cu content for the initial and annealed films. The sulfide sulfur ($S^{2-}$) associated with the NC structure and thioglycolate associated with the NC surface ($S^{1-}$) are taken into account, which is the most suitable approach for representing the composition of TGA stabilized CZTS colloidal NCs according to our previous work [37]; Table S2: Vibrational modes of carboxylic groups detected in the middle-infrared spectra of $Cu_4$, $Cu_2$, and $Cu_{0.25}$ initial (i.e., heated at 120 °C) and annealed at 220 °C samples [63,64].

**Author Contributions:** Conceptualization, O.S. (Oleksandr Stroyuk) and V.D.; methodology, O.R.; investigation, O.S. (Oleksandr Selyshchev), N.M. and Y.H.; synthesis, O.S. (Oleksandr Selyshchev) and O.R.; writing—original draft preparation, V.D., O.S. (Oleksandr Selyshchev), and S.K.; writing—review and editing, D.R.T.Z.; project administration, D.R.T.Z., and V.D.; funding acquisition, D.R.T.Z., and V.D. All authors have read and agreed to the published version of the manuscript.

**Funding:** The work was partially supported by DFG (ZA 146/45-1), CRDF (FSA-20-66703-0), and NAS of Ukraine (N7/21-H). V.D. and S.K. thank the Visiting Scholar Program of TU Chemnitz for funding their research stay at TU Chemnitz. Y.H. is grateful to the Alexander von Humboldt Foundation for funding.

**Institutional Review Board Statement:** Not applicable.

**Informed Consent Statement:** Not applicable.

**Data Availability Statement:** The data presented in this study are available on request from the corresponding author.

**Conflicts of Interest:** The authors declare that they have no conflict of interest.

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
