# Peer review of "Copper-Content Dependent Structural and Electrical Properties of CZTS Films Formed by “Green” Colloidal Nanocrystals"

_electronicmat, doi:10.3390/electronicmat3010013_

Round 1

Reviewer 1 Report

CuxS peak was well presented in Raman spectroscopy. This is a new finding and worth to be reported.

Reviewer 2 Report

The manuscript entitled: „Copper-content dependent structural and electrical properties of CZTS films formed by "green" colloidal nanocrystals” reports a study of CZTS nanocrystal samples with different Cu content, which were characterized by Raman scattering, mid-infrared and near-infrared absorption, XPS, electrical conductivity, and conductive atomic force microscopy. In the submitted paper, there are presented some important results that can be useful for further improvement in CZTS based solar cells. Below are some comments the authors need to address before publication. Suggest minor revision.

1. Introduction section has lots of basics. The interest of kesterite CZTS in the field of photovoltaic devices is clear but the authors should highlight the originality and the relevance of this work. What are the main contributions? Is it the novelty of some experimental procedures? Or the improvements of the fabrication protocols giving rise to samples with optimized properties or higher reproducibility?

2. In Conclusions section there is lack of reference to the results obtained from optical absorption. I suggest to complete this.

Reviewer 3 Report

This manuscript deals with the green synthesis of CZTS NCs varying in copper content in aqueous medium. The electronic, electrical, optical, and vibrational properties of CZTS NCs were studied with respect to the variation in the copper content. The author stated that an increase of the electrical conductivity of the NC films upon annealing at 220 oC is explained by three factors: formation of a CuxS nanophase at the CZTS NC surface, partial removal of ligands, and improved structural perfection. The presence of the CuxS phase is concluded to be the determinant factor for the CZTS NC film conductivity. By using various instrumental analyses such as Raman Scattering, Mid/near-IR, XPS and cAFM, author studied the above said properties and discussed with respect to temperature (annealing) and inert and air atmosphere. The author did quantity of work and the results are discussed scientifically. Based on the novelty and achievement, this manuscript should be “ACCEPTED” to publish in your journal “Electronic Materials”. However, the following factors need to be clarified prior to publication.

  1. How did the author fix/select the annealing temperatures as 110 and 220 oC?
  2. In Fig. 2, why was the peak for free thiol (S0) also found to be varied after annealing? Will it be affected by the annealing process too?

In Fig. 4b and 4c legend, Cu 0.25 seems to be 0-25. 
